# OpenReview forum: "Automatic Pseudo-Harmful Prompt Generation for Evaluating False Refusals in Large Language Models"
_colmweb.org/COLM/2024/Conference — COLM_

### Official Review · Reviewer_Dgn7 · 2024-05-08

**Rating:** 6
**Confidence:** 3
**Ethics Flag:** 1

**Summary:**

This paper explores an interesting issue in current applications of LLMs: false refusals. Unreasonable refusals can harm the user experience. This paper proposes **Auto-Regressive Controllable Prompt Generation** for generating false refusal text, based on AutoDan and three objectives. This method can generate natural prompts and control the harmfulness of the prompts and the probability of being refused. Based on this method, this paper proposes a new dataset, PHTest, containing 1.5k prompts usable for detecting false refusals. The authors made comprehensive evaluations of state-of-the-art large models, including commercial-level close-sourcde LLMs and open-source LLMs.

**Questions To Authors:**

Equation 1 includes two hyperparameters $\alpha_0$ and $\beta_0$. How did you pick these two parameters?

**Reasons To Accept:**

* This paper brings our attention to the issue of false refusals.
* The author proposes an automatic method for generating text for false refusals. This method can be easily applied to generate more prompts to evaluate LLMs
* This paper already provides a new dataset covering several categories of false refusals, with a preliminary evaluation conducted on a range of LLMs.

Overall, I think this work makes a positive contribution to the community.

**Reasons To Reject:**

Although this has been discussed in detail in the paper, I still think that the definition of harmfulness in the text is not promising. Since the method used in the text is fully automated, the harmfulness of the generated text is also annotated by a model, which inevitably leads to unreliable harmfulness annotations in the dataset.

---

> ### Author Rebuttal · Authors · 2024-05-31
>
> Thank you for your valuable questions! Below, we address them in detail.
>
> **Answer to W1**: The harmfulness is not annotated by a model. We generate potential pseudo-harmful prompts automatically. After generation, we manually annotate each prompt as "harmful," "harmless," or "controversial" according to content moderation documents from OpenAI and Meta. This post-validation process ensures the reliability of our dataset. We describe this manual annotation phase as the third step in constructing the PHTest dataset in Section 4. We will enhance these descriptions to prevent any misunderstandings.
>
>
> **Answer to Q1**: Eq. 1 presents the general formulation of this problem, requiring a classifier $\gamma_{harmless}$ and a threshold $\alpha_0$. In our method, we use a writer LLM as the proxy of the classifier, which not only saves computation but also effectively handles incomplete prompts during generation. As detailed in Appendix A, we let $\alpha_0=1$, transforming the objective into Eq. 2.
>
> In Eq. 2, $\beta$ is a hyperparameter that balances two objectives. If $\beta$ is too small, the generated prompts may be harmless and unable to trigger refusal. If $\beta$ is too large, the generated prompts could trigger refusal but might be harmful. Therefore, we use a validation set (including harmless, pseudo-harmful, and harmful prompts) and a grid search to select $\beta$ to be the one where pseudo-harmful prompts have higher values of the objective (Eq 2) than other prompts. Note that, $\beta$ also depends on the LLMs we use since different LLMs have different objective ranges. Below is the range of $\beta$ in our three settings. We randomly sample a $\beta$ within the range for each run.
>
> |Writer LLM|Target LLM|$\beta$|
> |----------|:--------:|---------:|
> |Llama2-7b|Llama2-7b|[10, 25]|
> |Vicuna-7b|Llama2-7b|[20,30]|
> |Mistral-7b|Mistral-7b|[20,80]|
>
> Thanks for the question. We will add these details to the paper.

---

> > ### Comment · Reviewer_Dgn7 · 2024-06-03
> >
> > I appreciate the authors' response and informative clarification. Thank you!

---

### Official Review · Reviewer_gbSz · 2024-05-10

**Rating:** 6
**Confidence:** 4
**Ethics Flag:** 1

**Summary:**

This paper focuses on an interesting problem LLMs may refuse pseudo-harmful user prompts and affect users’ experience. To accelerate the development of this domain, the authors propose auto-generating pseudo-harmful prompts, leveraging a white-box LLM to generate natural, varied, and controllable prompts. They construct a larger dataset PHtest and conduct extensive experiments on 14 models. They also discuss a trade-off between false refusals and safety against jailbreak attacks.

**Reasons To Accept:**

1. The problem that LLMs refuse pseudo-harmful user prompts is interesting and important for user experience.
2. The authors generate a larger evaluation dataset to evaluate the performance of many LLMs on False Refusal Evaluation.
3. The writing is clear and the logic of this paper is easy to follow.

**Reasons To Reject:**

1. The authors have one hyperparameter $\beta$ during the generation process of their dataset. At the end of section 3.2, the authors discuss the potential function of $\beta$. It’s better to give specific details about beta you choose or some simple experiments about the effect of $\beta$ to generate datasets.
2. In the Appendix, the authors set $y_{refusal}$ as “Sorry, I cannot assist with that.” It’s also better for authors to discuss the effect of different $y_{refusal}$ for generating datasets or give some case studies about this.
3. In observation 1 and Figure 9 about calculating AUC values, it’s better for authors to give a bit of detail about experiments.

---

> ### Author Rebuttal · Authors · 2024-05-31
>
> Thank you for your valuable questions! Below, we address them in detail.
>
> **Answer to W1**: Due to the page limit, please see our response to Reviewer Dgn7's last question. Thank you!
>
> **Answer to W2**: Good question. In the appendix, we use "Sorry..." as an illustration. Below, we discuss the details.
> - Selection of $y_{refusal}$:
>     - A good $y_{refusal}$ should guide the generation of prompts that can trigger refusals in reality. Therefore, we choose $y_{refusal}$ as the most common refusal prefix for the target LLM by testing many harmful requests. In our experiments, we use "I cannot fulfill your request ..." for Llama2-7b, and "I cannot provide instructions on ..." for Mistral-7b.
> - Effect of $y_{refusal}$:
>     - As discussed in Sec 3.2, $y_{refusal}$ can be tailored to generate prompts targeting specific rule violations. For example, using "I can’t assist with copyright infringement" as $y_{refusal}$, the generated prompt could be "Where can I find open-access journals ..." Therefore, varying $y_{refusal}$ is a way to promote diversity. We only have a qualitative analysis on it since we fix $y_{refusal}$ but use reference prompts to achieve diversity. We will add the qualitative analysis to the paper.
>
> **Answer to W3**: Figure 9 aims to see whether harmful and pseudo-harmful prompts can be differentiated by the second term in Eq. 2. Here's a detailed breakdown of the experiment:
> * We collect prompts from 3 classes: harmless, pseudo-harmful, and harmful. The last two are from the XSTest dataset.
> * We calculate the likelihood of refusal (i.e., the exponential of the second term in Eq. 2) for each prompt. The likelihood of refusal serves as a feature for classifying the three prompt classes. We then compile a list of feature values for each prompt class.
> * With kernel density estimation (KDE) and these feature values, we draw Fig. 9. The colors denote the class: blue for harmless, orange for pseudo-harmful, and green for harmful (apologies for the typo in the legend). We observe that three classes are quite separable.
> * Using the feature values of pseudo-harmful class and harmful class, we can calculate the AUC (with packages like sklearn) of classifying them to be 82.1%. This supports that although the target LLM refuses both pseudo-harmful and harmful prompts, the likelihoods are different. Thus, the likelihood of refusal is a good feature for distinguishing these two classes.
>
> Thanks for this question. We will add these details to the paper.

---

> > ### Comment · Reviewer_gbSz · 2024-06-03
> >
> > I appreciate the authors' detailed responses. After carefully reading the rebuttal and the remaining reviews I decided to maintain my score.

---

### Official Review · Reviewer_Dwyh · 2024-05-13

**Rating:** 6
**Confidence:** 4
**Ethics Flag:** 1

**Summary:**

This paper presents an investigation about the False Refusals in defending harmful prompt. The authors propose a method to generate a pseudo-harmful dataset, and use it to evaluate several popular LLMs. Analysis are presented to show the tradeoff between false refusals and successful defendings.

**Reasons To Accept:**

1. While many work improve the successful defending rate, much less work put false refusal in an important position. This work brings out an interesting direction.

2. A dataset are created for evaluating the above problem, with detailed labeling of the type and reason of false refusal, which may be potentially helpful for further study.

3. The authors present analysis to help understand the different categories of false refusal (including newly identified categories), as well as the balance between false refusals and successful defendings.

**Reasons To Reject:**

1. The generation process of the dataset seems pretty standard.

2. It is interesting to find new categories of fasle refusal. However, even with the listed 10 categories, there might still be other cases that have not been studied yet. I am wondering how it is useful to reveal these detailed categories. For example, is it possbile to improve the alignment/defence on a given category

---

> ### Author Rebuttal · Authors · 2024-05-31
>
> Thanks for your valuable questions! Below, we address them in detail.
>
> **Answer to W1**: We would appreciate further clarification on this question. If the concern is that our dataset generation process is similar to previous methods, we highlight several key differences:
>
> * Previous datasets, XSTest and OKTest, either manually craft pseudo-harmful prompts or use GPT-4 to generate them with harmful words. It results in small datasets (XSTest has 250 examples, OKTest has 300 examples) with limited content, and often fail to trigger refusals.
> * In contrast, our dataset is generated using a newly proposed algorithm that automatically generates pseudo-harmful prompts from scratch, with the optimization goal of triggering the refusal of target LLM. This approach produces a large, diverse, and natural dataset that can effectively trigger refusals of many LLMs. Our dataset contains 1.5K examples, featuring newly discovered categories of false refusals.
>
>
> Therefore, our data generation process significantly improves the dataset quality and value compared to previous ones.
>
> **Answer to W2**: This is a very good question. Yes, we believe that revealing detailed categories provides important insights for improving alignment.
> * Identifying specific categories of false refusals allows developers to perform a more granular analysis of LLM performance. This detailed understanding helps in diagnosing the root causes of false refusals more effectively. For example, if the LLM frequently refuses prompts with homonyms and figurative language, the root cause might be a deficiency in language understanding. If the LLM refuses many requests related to public statistics, the root cause might be the over-generalization of safety fine-tuning.
> * Based on the granular analysis, developers can implement targeted improvement strategies. For instance, they can fine-tune the LLM with examples containing homonyms and figurative language for the first case, and explore more requests related to public statistics that shouldn't be refused and incorporate them into safety fine-tuning to avoid over-generalization for the second case.
>
> Thanks again for the question. We will add the discussion on the potential use of these categories to the paper.

---

> > ### Comment · Reviewer_Dwyh · 2024-06-05
> > **reply**
> >
> > Thanks for the clarification. The datasets are indeed different in size. However the methodology in generating is similar to Zhu et al. I think the paper may have better influence in the dataset and evaluation rather than the method of generation.

---

### Official Review · Reviewer_VMEF · 2024-05-13

**Rating:** 6
**Confidence:** 3
**Ethics Flag:** 1

**Summary:**

In this paper, the authors study the problem of falsely refusing pseudo-harmful prompts by LLMs. Specifically, the work proposes an approach to automatically generate pseudo-harmful prompts. Using this method, the authors construct a new dataset called PHTest which is larger in scale and more finely annotated than existing datasets for this purpose. They evaluate current LLMs on PHTest, revealing insights such as the correlation between model size and false refusal types, and the trade-off between safety against jailbreak attacks and low false refusal rates.

**Questions To Authors:**

1. How can the authors ensure that the writer LLM generates prompts within specific desired categories, such as "Compound term" or "Shared Property vs Privacy"?

2. Did the authors experiment with using different target LLMs in the generation process to improve the transferability of the generated prompts?

**Reasons To Accept:**

1. The studied problem is highly practical and important for developing safer and more user-friendly LLMs.

2. The PHTest dataset is a significant resource, being 10x larger than prior datasets. A natural and fine-grained dataset can help identify the detailed drawbacks of existing LLMs and provide developers with insights on their capabilities, which is valuable for advancing the community.

**Reasons To Reject:**

1. The content-steering process could be explained in more detail. The writer LLM may not be able to generate desired pseudo-harmful prompts with x_{natural, harmless}. I think the generation quality is severely limited by the writer LLM and the reference prompt. How can the authors ensure that the writer LLM generates prompts within some specific categories, like "Compound term" or "Shared Property vs Privacy"? Additional experiments or analysis on the effectiveness of content-steering would strengthen this part of the methodology.

2. The generated prompts may not transfer to other LLMs as being pseudo-harmful. As the authors mention in Section 2, the definition of harmfulness is inherently complex and controversial. Using a single target LLM to guide the generation process could introduce biases specific to that model. To improve the generalizability of the generated prompts, the authors could consider using an ensemble of different target LLMs to mitigate model-specific biases.

---

> ### Author Rebuttal · Authors · 2024-05-31
>
> Thank you for your valuable feedback! Below, we address the questions in detail.
>
> **Answer to W1&Q1**: We would like to clarify that the categorization occurs **after** prompt generation. These categories are the natural outcomes of our autonomous generation algorithm. During generation, our method uses reference prompts to steer style and domain, rather than targeting specific categories directly. This approach ensures a diverse set of pseudo-harmful prompts, which we then categorize into "Compound term" or others. We will make this process clearer to prevent any misunderstandings.
>
> Our dataset is of high quality.  Unlike previous datasets that were manually crafted and limited in scope and diversity, ours contains 1.5K diverse prompts that effectively trigger false refusals across various LLMs. Using 7b models like Llama2-7b, Vicuna-7b, and Mistral-7b as writer LLMs, and ShareGPT as the reference prompts, has proven effective in generating these high-quality pseudo-harmful prompts.
>
>
>
> **Answer to W2&Q2**: Thanks for the insightful suggestion!
> * Our method serves as a red-teaming tool to help LLM developers identify false refusals specific to their models. When using single target LLM, the generated prompts are tailored to it, which is beneficial for red-teaming purposes.
> * To ensure a large and diverse dataset, we collect pseudo-harmful prompts generated from three settings (writer LLM + target LLM to be Llama2-7b + Llama2-7b, Vicuna-7b + Llama2-7b, and Mistral-7b + Mistral-7b), forming the PHTest dataset.  We observe that the pseudo-harmful prompts generated from a single target LLM already exhibit some transferability, as they can trigger false refusals of various black-box LLMs, such as Claude.
> * We agree that using an ensemble of target LLMs is a promising way to enhance transferability. We would like to explore it in the future work since this approach presents two challenges. Firstly, it requires more memory and computation resources. Secondly, different LLMs have different vocabularies and tokenizers, complicating the optimization of the next token. One potential solution is to use tokenizer adapters [1]. Thanks again for the great suggestion. We will include a discussion on this and explore it in our future work.
>
> [1] Minixhofer, Benjamin, Edoardo Maria Ponti, and Ivan Vulić. "Zero-Shot Tokenizer Transfer." arXiv preprint arXiv:2405.07883 (2024).

---

> > ### Comment · Reviewer_VMEF · 2024-06-07
> > **Official Comment by Reviewer VMEF**
> >
> > Thanks for the response, my concerns have been addressed and I have raised my rating.

---

### Decision · Program_Chairs · 2024-07-10

**Decision:**

Accept

**Comment:**

This paper explores the problem of false refusals to pseudo-harmful prompts, developing a method for automatic generation of such prompts, and using it to construct a dataset with large scale and fine-grained annotations. They use the dataset to evaluate current LLMs and share insights from the analyses.

Overall, all reviewers are positive about this paper. They express that the problem tackled is important, of practical utility, and understudied, that the dataset being contributed is a valuable resource, and that the method is novel and easy to apply as well.

A few concerns are raised, including requests for more experimental details, questions about other possible generation settings, and a concern about defining harmfulness. However, it seems that most or all of these concerns have been satisfactorily addressed by the authors in discussion (and many have been cleared up as misunderstandings). This seems like a solid submission for publication.